# Postpartum Depression after Maternal Isolation during the COVID-19 Pandemic: The MUMI-19 Study (Mothers Undergoing Mental Impact of COVID-19 Pandemic)

**DOI:** 10.3390/jcm11195504

**Published:** 2022-09-20

**Authors:** Lina Boudiaf, Françoise Dupont, Christèle Gras-Le Guen, Anne Sauvaget, Maxime Leroy, Thibault Thubert, Norbert Winer, Vincent Dochez

**Affiliations:** 1Service de Gynécologie-Obstétrique, CHU de Nantes, CEDEX, 44093 Nantes, France; 2Service de Pédopsychiatrie, CHU de Nantes, CEDEX, 44093 Nantes, France; 3Service de Pédiatrie, CHU de Nantes, CEDEX, 44093 Nantes, France; 4Laboratoire Mouvement, Interactions, Performance (MIP), EA 4334, Nantes Université, CEDEX, 44322 Nantes, France; 5Plateforme de Méthodologie et Biostatistiques, CHU de Nantes, CEDEX, 44093 Nantes, France; 6Centre d’Investigation Clinique (CIC), CHU de Nantes, CEDEX, 44093 Nantes, France

**Keywords:** COVID-19, depressive disorders, postpartum depression, Edinburgh Postnatal Depression Scale, postpartum blues

## Abstract

Background: The COVID-19 pandemic has shaken the world by imposing unprecedented health measures, including in the postpartum period. Objectives: We aim to assess the impact of maternal isolation in the immediate postpartum period on the rate of postpartum depression (PPD) in a tertiary center. Study Design: We conducted a prospective cohort study, between 22 April and 29 October 2020, using anonymous questionnaires on 265 participants (129 during lockdown and 135 outside). The Edinburgh Postnatal Depression Scale (EPDS) was used as screening for PPD. We used a univariate logistic regression model to analyze the association between risk factors and PPD. Results: There was no difference between the two groups for PPD assessed by an EPDS score >10.5 on day 30 and/or day 60 (23.1% vs. 29.3%, *p* = 0.661) but on day 3 it was higher (31% vs. 17.8%, *p* = 0.015) during the lockdown period and partners were more impacted psychologically (48.3% vs. 10.5%, *p* < 0.001). Parity ≥1 was a protective factor for PPD (OR = 0.2, 95% CI [0.1–0.6], *p* = 0.003). Risk factors of PPD were: history of psychological abuses (OR = 6.4, CI 95% [1.1–37.6], *p* = 0.04), stressful life event (OR = 4.5, CI 95% [1.6–12.6], *p* = 0.004), and bad birth experience (OR = 5.1, CI 95% [1.4–17.8], *p* = 0.012). Conclusion: Maternal isolation in the immediate postpartum period is associated with an increased rate of moderate to severe symptoms of postpartum blues. The well-known long-term consequences of PPD must be balanced against the expected benefits of partner’s restrictive access to maternity ward.

## 1. Background

The coronavirus pandemic has shaken the world since its emergence in Wuhan in December 2019 [1] and has rapidly grown as a new global public health issue. Encouraged by the World Health Organization (WHO) [2], various governments quickly set up essential protective measures to slow down the formation of new contagion focuses and preserve the reception capacities of their hospitals. A first lockdown was established in France from 17 March to 11 May 2020, in parallel with a tightening of access to care facilities for those accompanying patients. As of 27 March, the Collège National des Gynécologues et Obstétriciens de France (CNGOF) proposed that partners of new parturient should be accepted in the maternity hospital only for the delivery. Visits were strictly forbidden during the entire stay in the maternity ward, forcing mothers to cope alone with the beginnings of their parenthood in a period of psychological vulnerability, with women more likely to develop depression [3].

Postpartum depression (PPD) is thought to affect about 10–20% of mothers [4,5] and suicide is responsible for 20% of postpartum deaths [6]. Due to heterogeneity in screening and diagnostic methods, cultural variations [7] and the standard of living in the country studied [8], it is often under-screened [9,10]. It is diagnosed according to the Diagnostic and Statistical Manual of Mental Disorders 5 (DSM 5) criteria. Unlike the common third-day postpartum blues (PPB) observed in 30–80% women, and whose symptoms resolve spontaneously within a week, PPD symptoms appear before the sixth week postpartum and generally resolve within a year [11].

To date, the literature is poor on the psychological impact of these new isolation measures in the immediate postpartum period. Our work is all the more justified as we cannot exclude new periods of restrictions and barrier measures, for which the consequences on women’s mental health are still unknown and unpredictable. The aim of our study is to evaluate the impact of maternal isolation in the immediate postpartum period on the rate of PPD in a tertiary center. In addition, we wish to assess its impact on EPDS scale at day 3, 30 and 60 of postpartum, on the experience of the maternity stay, on breastfeeding, on return home, and the mental impact on the partner.

## 2. Material and Methods

We conducted a prospective cohort study at the maternity ward of the University Hospital of Nantes, which had 4380 births in 2019. We included patients aged 18 to 43 years (mean = 30.9 years), able to read and write French, during the period from 22 April to 29 October 2020. All patients were recruited after delivery. Of the 352 patients initially invited to participate in the study, 264 were included, 129 during lockdown and 135 during the post lockdown period. We will compare 2 groups: the "lockdown" group or ld group (during the lockdown period from 22 April to 11 May 2020) and the "post lockdown" group or pld group (between 12 May and 29 October 2020). There was no difference between the two periods, in terms of standard operating procedures, staffing condition and healthcare capacity and resources in the hospital. The working conditions were strictly similar between the 2 groups.

Patients who refused to participate, had a medical condition (decompensated hypothyroidism, neurodegenerative disorders) or significant substance use that could falsely induce PPD symptoms or single mothers were excluded. During the lockdown period, partners were required to leave the maternity ward immediately after delivery and could not visit their partners. Data were collected using standardized anonymous questionnaires. The Edinburgh Postnatal Depression Scale (EPDS), the most commonly used scale for PPD screening [12], was used in our study, with a French cut-off score of 10.5, providing a sensitivity of 92% and a specificity of 80% [13]. Its results are systematically a whole number. A score >10.5 or more indicates possible depression. However, clinical confirmation is required to make a diagnosis (36–47% of women with this score do not have a confirmed diagnosis of depression). In the postpartum period, a score greater than or equal to 12 corresponds to a 60% risk of suffering from depression, whereas a score lower than 10.5 indicates a 96% chance of not suffering from depression. We used the cut-off of 10.5 as in other studies in the literature as the primary endpoint. We also analyzed as secondary endpoint the cut-off of 13 to detect a severe characterized depressive episode [14].

During the first contact, women received an explanation on the aim of the study, that is to say to assess the impact of their isolation during hospitalization on their mental state. Modalities were explained in detail. Once consent was obtained, an inclusion number was assigned to allow anonymization of the data. Each patient received 3 paper questionnaires. The first one had to be collected between the 2nd and the 7th day postpartum. It was organized in 4 sections: EPDS, the search for risk factors for PPD established in the literature (Table 1, Appendix A), the feeling during the stay in the postpartum period, and a final part concerning the return to home, the feeling of their partner and breastfeeding.

The PPD score was most often established before discharge. The knowledge of a positive score made it possible to supervise the return at home by proposing the implementation of psychological support. Patients with a positive score on the last question of the EPDS were systematically referred for psychiatric evaluation. The patient left the maternity hospital with two identic follow-up questionnaires, to be sent back 30 and 60 days after the birth on a secure e-mail box. They consisted of the EPDS and 4 questions on breastfeeding and home return. When a score >10.5 was discovered afterwards, the patient was contacted by telephone by one of the two investigators to propose psychological follow-up and a short course of treatment was set up if necessary. Our primary endpoint was a combined variable that measured the rate of patients with an EPDS score >10.5 at day 30 and/or day 60 in both groups. The patient was considered to have PPD when she had a score >10.5 on at least one or both questionnaires, and a moderate to severe postpartum blues (PPB) with a score >10.5 on the EPDS at day 3. We received ethical approval from the Groupe Nantais d’Éthique dans le Domaine de la Santé (GNEDS) on 22 April 2020.

Statistically, comparisons between groups for continuous variables were made using the Mann–Whitney–Wilcoxon test, which is a non-parametric test, rather than the t-test, because the data did not meet the requirements of parametric tests. Comparisons between groups were made using Chi-squared tests or Fisher exact tests if the conditions were not met for categorical variables. Continuous variables were described by their mean and standard deviation; categorical variables were described by their associated number and percentage. Univariate logistic regressions were used to estimate the odds ratios of the different risk factors. For the study of secondary endpoints, we performed subgroup analyses according to the EPDS score at day 30 and/or 60 in each group. The analyses were performed with R software (v4.0). We used STROBE statement for the redaction of our study [15].

## 3. Results

Of the 352 patients initially invited to participate in the study, 264 were included, 129 during the lockdown period from 22 April to 11 May 2020 (the “lockdown” group or ld group) and 135 afterwards (the “post lockdown” group or pld group) (Figure 1). The global follow-up rate at day 30 was 34.4% with 40.3% during the ld period vs. 28.9% in the post lockdown period (pld period). It was 23.4% at day 60, with 24% during the ld period vs. 17.8% outside.

The demographic characteristics of the patients are described in Table 1.

Combining the response rates to the questionnaires at day 30 and day 60, 97 patients completed at least one of the follow-up questionnaires, 56 in ld group, versus 41 in the pld group. Of these, 25.7% had an EPDS score >10.5. However, there was no significant difference between the two groups (23.2% vs. 29.3%, *p* = 0.661), (Table 2).

There was a significant difference between the two groups in the proportion of patients with a positive EPDS score at D3 (31% vs. 17.8%; *p* = 0.015). Of these, 57.8% (37/64) had a score ≥ 13 considered potentially severe, and specifically 62.5% (25/40) in the ld group.

At day 30 and day 60 independently, patients in the ld group were no more likely to have an EPDS score >10.5 (21.2% vs. 23.1%, *p* = 0.999 and 12.9% vs. 16.7%, *p* = 0.718 respectively).

We know that a total of 26 patients were referred for psychological evaluation (16 in ld group vs. 10 pld group). Of these 26, 19 patients actually met with the department psychologist.

There was no difference in the rate of EPDS score >10.5 or 13 between patients who completed the questionnaire at D3 and those who completed at D30 and/or D60 (Figure 2). There was no difference in the rate of EPDS score >10.5 or 13 between patients who completed the questionnaire at D3 and those who completed at D30 and/or D60. Of the 64 patients with a score >10.5 at D3, 17 (26.6%) responded at D30 and 15 (23.4%) at D60, compared with 74 (37.0%) patients at D30 among the 200 who had a score <10.5, and 40 (20.0%) at D60.

At day 3, the ld women were more likely to report that restrictive measures brought challenges their early maternity experience than those in the pld group (*p* < 0.001), (Table 3), but also at D30 and day 60 (48.2% vs. 2.5%, *p* < 0.001), (Appendix A). At day 3, the ld women felt less able to care for their child than those in the pld group (24% vs. 4.7%, *p* < 0.001).

Due to the absence of the partner in the postpartum period, 17.6% of the ld patients stated at day 3 that they feared that there would be a mismatch with the partner in the way the baby was cared for at home, compared with 5.5% in the pld group (*p* = 0.05) (Table 3). This discrepancy was actually experienced by the patients at follow-up (20.7% vs. 0%, *p* = 0.028) (Appendix A). Women in the ld group undertook more early discharges than those in the pld group (43.7% vs. 20.5%, *p* < 0.001) (Table 3), although patients in the ld group with PPD had more difficulty undertaking them (8.3% vs. 53.7%, *p* = 0.007), (Appendix A).

The breastfeeding rate at day 3 postpartum was 81.6% and we found no significant difference between the two groups (*p* = 0.638) (Table 3). In the subgroup analysis, patients in the ld group with PPD had more difficult breastfeeding in the first week postpartum (50% vs. 18.3%, *p* = 0.003), (Appendix A). The breastfeeding was maintained at two months’ follow-up in the whole cohort, 48.2% (27/56) in the ld group and 30.0% (12/40) in the pld group (Appendix A).

At day 3, the partners in the ld group seemed to have been significantly more psychologically affected by the separation (48.3% vs. 10.5%, *p* < 0.001), (Table 3), which was also true at follow-up (46.4% vs. 5.1%, *p* < 0.001), (Appendix A).

Except for the obstetric history, unwanted pregnancy and unwanted absence of partner at delivery, the groups were comparable on all risk factors (Table 1). In subgroup analysis in each of the two groups, we wanted to know whether a positive EPDS score on at least one of the two follow-up questionnaires was more associated with the presence of risk factors. Thus, in the confined group, patients with a score >10.5 were more likely to be nulliparous (92.3% vs. 39.5%, *p* = 0.001), or to have had a traumatic birth experience (38.5% vs. 11.9%, *p* = 0.045). In the pld group, patients with a score >10.5 had significantly experience a stressful event during pregnancy (91.7% vs. 46.4%, *p* = 0.012), (Appendix A).

Based on a positive score at day 30 and/or day 60, in the overall cohort, only parity ≥ 1 was a protective factor for PPD, with a decreased risk of 80% (OR = 0.2, *p* = 0.003, 95% CI [0.1–0.6]). Three parameters could be highlighted as risk factors for PPD: history of psychological abuse (OR = 6.4 [1.1–37.6], *p* = 0.04), presence of stress during pregnancy (OR = 4.5, 95% CI [1.6–12.6], *p* = 0.004) and traumatic birth experience (OR = 5.1, 95% CI [1.4–17.8], *p* = 0.012) (Table 4).

## 4. Discussion

To our knowledge, this is the first study to specifically address the issue of maternal isolation in the immediate postpartum period and its psychological impact on the mother, although the literature on the broader issue of psychology in postpartum during the COVID-19 pandemic has since expanded.

We did not find a significant difference between our two groups regarding PPD (23.1% vs. 29.3%, *p* = 0.661). Recent studies tend to support an increased risk of PPD in women who gave birth during confinement [16].

The main difference observed was regarding the proportion of scores >10.5 at D3 in the ld group (31% vs. 17.8%, *p* = 0.015), reflecting moderate to severe postpartum blues (PPB) symptoms. We found that parity ≥1 was a protective factor for PPD (OR = 0.2, 95% CI [0.1–0.6], *p* = 0.003) and psychological abuse was correlated with a risk of PPD (OR = 6.4, 95% CI [1.1–37.6], *p* = 0.004).

Partners in the confined group appeared to be more psychologically affected (48.3% vs. 10.5%, *p* < 0.001) although we did not specifically survey partners to support our findings.

In a cross-sectional Chinese cohort study of 864 women using the EPDS with a cut-off score of 10 [17], the prevalence of PPD between 1.5 and 3 months postpartum was shown to be 30%. However, there was no control group to assess the real impact of confinement on PPD rate and the lower EPDS threshold may have increased the number of patients screened compared to our study. Furthermore, none of these studies specified whether or not the patient was accompanied by her partner during the postpartum period, making comparison with our study difficult since we primarily aimed to evaluate the impact of maternal isolation on the PPD rate, not only the impact of confinement on PPD rate.

In a recent meta-analysis [18] including 24246 patients (20 569 during ld and 3677 outside), two studies assessing PPB using the EPDS at D2 reported a prevalence of 18% positive score (95% CI [14–23%], I2 = 85.2%). Among them, Oskovi-Kaplan and al. in a Turkish cohort study [19] of 223 women, taking a score ≥ 13, showed that 14.7% of women were at risk of PPD. This result is close to the prevalence of 19.4% observed in our study with a score ≥ 13, although the exclusion of women with medical, psychiatric or obstetric pathology may explain a lower prevalence in their cohort. Independently of the absence of the partner, we can formulate several hypotheses to explain the significant difference observed between our two groups at D3. Firstly, the recruitment of our pld group started the day after the lifting of the restriction of access to partners to the maternity ward, which could generate a pld "honeymoon" period. Thus, we observed that many patients felt satisfied having avoided those restrictions. This probably put their symptoms into perspective. This hypothesis is corroborated by the results of the CoviPrev survey [20]. It is interesting that the women in the ld group were more likely to report that restrictive measures brought challenges to their early maternity experiences at day 30 and 60 as well, but there were no differences in their EPDS scores at these times. This may give some rationale that, despite not being reflected in reported symptoms of depression, these women still may need additional support that may be missed in postpartum screenings.

The literature is non-existent on paternal postpartum depression during the COVID-19 pandemic. In a 2015 meta-analysis of 74 studies from 1980 to 2015 including 41,480 participants from 22 different countries [21], the prevalence of paternal postpartum depression was 7.8% in the first three months (95% CI [6.3–9.7%]), similar to our findings at two months in the pld group (5.1%). The maternal depression was found to be associated with the development of paternal depression (*p* < 0.001), which is corroborated in Goodman’s study [22], which identified it as the main predictor of PPPD. Indeed, this review of 20 studies showed that paternal depression up to one year postpartum ranged from 1.2% to 25.5%, and from 24% to 50% in men whose wives had PPD, confirming the strong thymic interconnection between the two members of the couple [23]. This was confirmed in our cohort, where there was a joint increase in depressive symptoms in patients and their partners in the first week postpartum.

The average of breastfeeding was 81.6% in the first week postpartum in our cohort, which is higher than the national one (65.7% in 2017). It is known that younger women (<25 years), with poor social support or with low income [24,25] breastfeed statistically less frequently and for less time. Our population did not present those risk factors. Furthermore, our study took place in a tertiary maternity hospital, where women were made aware of the benefits of breastfeeding and where childbirth preparation classes were systematically offered [26]. The breastfeeding was maintained at two months’ follow-up: 48.2% (27/56) in the ld group and 30.0% (12/40) in the pld group. This appears to be higher in the ld group. This could be explained by the fact that patients were more available to breastfeed as they were less disturbed by visits, but also by the fact that patients only received advice from health professionals and not from family members or acquaintances.

## 5. Strengths and Limitations

Among the limitations of our study, it is a monocentric study. The clear explanation of the purpose of the study at the time of inclusion may have induced a selection bias, recruiting more people for whom concerns about isolation had a greater emotional value. On a theoretical level, this may have overestimated the risk of PPD in both groups. The main limitation is about the significant number of patients lost to follow-up. This high attrition proportion entails a high likelihood that this would tend to cause selection bias. In addition, the resulting small numbers could give rise to the problem of insufficient power. Therefore, this may have impacted our primary endpoint, which was based exclusively on our follow-up data.

It is the first study to specifically address the issue of maternal isolation in the immediate postpartum period and its psychological impact on the mother. Furthermore, we had a controlled group, and the two groups were comparable on almost all risk factors, leading us to limit selection bias. The data for our primary and secondary outcomes were collected prospectively, which is a crucial asset for the reliability of the information collected and its interpretation.

In this new context with major public health issues, some French multicenter prospective studies are underway and may support our initial hypotheses [27].

## 6. Conclusions

Due to the unprecedented health measures imposed in the context of the COVID-19 pandemic, maternal isolation in the immediate postpartum period is associated with an increased risk of developing more moderate to severe postpartum blues, putting women at greater risk of postpartum depression, especially if they are nulliparous, have a history of psychological abuse, per-partum stress, or have had a poor birth experience. Our study provides an initial overview of the psychological repercussions of this unexpected situation.

## Figures and Tables

**Figure 1 jcm-11-05504-f001:**
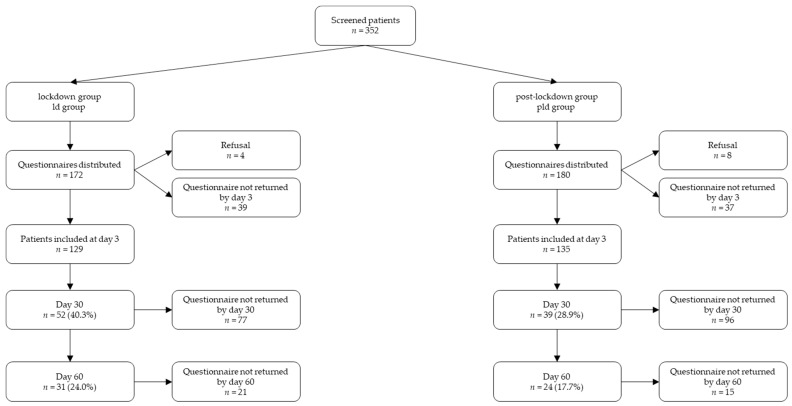
Flow chart.

**Figure 2 jcm-11-05504-f002:**
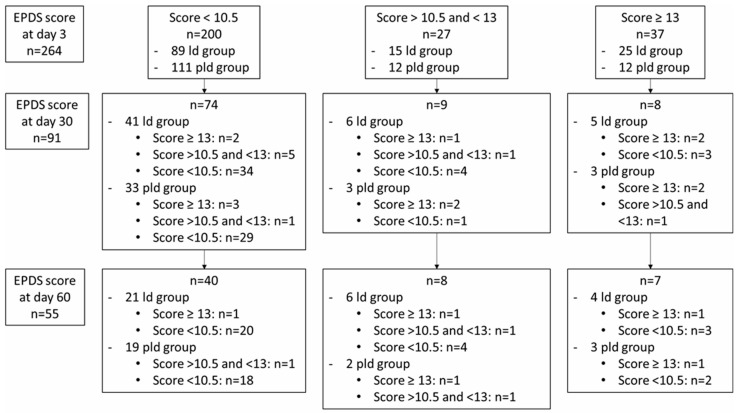
EPDS score by follow-up and questionnaire completion at D30 and/or D60. EPDS = Edinburgh Postnatal Depression Scale, ld = lockdown, pld = post lockdown.

**Table 1 jcm-11-05504-t001:** Risk factors of Postpartum Depression.

	Total	Lockdown Group (ld Group)	Post Lockdown Group (pld Group)	*p*
	(*n* = 264)	(*n* = 129)	(*n* = 135)	
Age (years)	*n* = 255	*n* = 126	*n* = 129	
Mean age ± SD	30.9 (±5.5)	30.8 (±5.5)	30.9 (±5.5)	0.953
≤24	38 (14.9%)	18 (14.3%)	20 (15.5%)	
25–29	61 (23.9%)	32 (25.4%)	29 (22.5%)	
30–34	85 (33.3%)	41 (32.5%)	44 (34.1%)	
>34	71 (27.8%)	35 (27.8%)	36 (27.9%)	
Parity	*n* = 254	*n* = 126	*n* = 128	
Nulliparity	125 (49.2%)	67 (53.2%)	58 (45.3%)	0.259
Parity ≥ 1	129 (50.8%)	59 (46.8%)	70 (54.7%)	
History of violence during childhood	*n* = 258	*n* = 126	*n* = 132	0.240
Psychological	15 (5.8%)	11 (8.7%)	4 (3.0%)	
Physical	8 (3.1%)	3 (2.4%)	5 (3.8%)	
Physical and psychological	8 (3.1%)	4 (3.2%)	4 (3.0%)	
History of medical affection impacting pregnancy	49/254 (19.3%)	22/126 (17.5%)	27/128 (21.1%)	0.565
Presence of psychiatric history	17/257 (6.6%)	7/125 (5.6%)	10/132 (7.6%)	0.700
Presence of familial psychiatric history	27/258 (10.5%)	14/126 (11.1%)	13/132 (9.8%)	0.898
Presence of a difficult obstetrical history	29/254 (11.4%)	8/126 (6.3%)	21/128 (16.4%)	0.020
Precarious situation	13/258 (5.0%)	9/126 (7.1%)	4/132 (3.0%)	0.221
Homeless patient	9/258 (3.5%)	5/126 (4.0%)	4/133 (3.0%)	0.744
Presence of marital difficulties	10/257 (3.9%)	6/126 (4.8%)	4/131 (3.1%)	0.534
Insufficient patient social support	5/255 (2.0%)	4/122 (3.3%)	1/133 (0.8%)	0.196
Existence of a migratory pathway for the patient	27/250 (10.8%)	17/122 (13.9%)	10/128 (7.8%)	0.175
Toxic use	30/259 (11.6%)	16/126 (12.7%)	14/133 (10.5%)	0.725
Current unwanted pregnancy	22/260 (8.5%)	16/127 (12.6%)	6/133 (4.5%)	0.034
Pregnancy difficult to obtain	37/230 (16.1%)	18/107 (16.8%)	19/133 (15.4%)	0.918
Pregnancy difficult to achieve	89/251 (35.5%)	44/123 (35.8%)	45/128 (35.2%)	0.999
Presence of per-partum stress	107/250 (42.8%)	50/120 (41.7%)	57/130 (43.8%)	0.826
Type of delivery	*n* = 252	*n* = 126	*n* = 128	0.970
Spontaneous vaginal delivery	164 (64.6%)	82 (65.1%)	82 (64.1%)	
Vacuum extraction	22 (8.7%)	8 (6.3%)	14 (10.9%)	
Forceps	4 (1.6%)	2 (1.6%)	2 (1.6%)	
Spatulas	10 (3.9%)	6 (4.8%)	4 (3.1%)	
C-section	52 (20.5%)	26 (20.6%)	26 (20.3%)	
Complications after delivery	*n* = 259	*n* = 129	*n* = 130	0.438
Preterm birth	8 (3.1%)	3 (2.3%)	5 (3.8%)	
Obstetrical anal sphincter injury (stage 3 or 4)	1 (0.4%)	0	1 (0.8%)	
Severe delivery hemorrhage > 1000 mL	7 (2.7%)	2 (1.6%)	5 (3.8%)	
Uterine rupture	1 (0.4%)	1 (0.8%)	0	
Traumatic experience of childbirth	32/254 (12.6%)	19/123 (15.4%)	13/131 (9.9%)	0.256
Absence of partner at delivery	*n* = 256	*n* = 125	*n* = 132	0.004
Unwanted absence	27 (10.5%)	18 (14.5%)	9 (6.8%)	
Desired absence	9 (3.5%)	8 (6.5%)	1 (0.8%)	
Mother/child separation at birth	20/255 (7.8%)	10/125 (8.1%)	10/132 (7.6%)	0.999
Length of stay in the maternity	*n* = 192	*n* = 90	*n* = 102	0.993
Mean length (d ± SD)	3.8 (±1.5)	3.7 (±1.1)	3.9 (±1.8)	
>3 days	97 (50.5%)	46/90 (51.1%)	51/102 (50.0%)	

**Table 2 jcm-11-05504-t002:** EPDS results in the two groups.

	Lockdown Group (ld Group)	Post Lockdown Group (pld Group)	*p*
EPDS at day 3	*n* = 129	*n* = 135	0.015
EPDS < 10.5	89 (69.0%)	111 (82.2%)	
EPDS > 10.5	40 (31.0%)	24 (17.8%)	
EPDS ≥ 13	25 (19.4%)	12 (8.9%)	
EPDS at day 30	*n* = 52	*n* = 39	0.999
EPDS < 10.5	41 (78.8%)	30 (76.9%)	
EPDS > 10.5	11 (21.2%)	9 (33.1%)	
EPDS ≥ 13	5 (9.6%)	7 (17.9%)	
EPDS at day 60	*n* = 31	*n* = 24	0.718
EPDS < 10.5	27 (87.1%)	20 (83.3%)	
EPDS > 10.5	4 (12.9%)	4 (16.7%)	
EPDS ≥ 13	3 (9.7%)	2 (8.3%)	
Combined variable EPDS at day 30 and/or day 60	*n* = 56	*n* = 41	0.661
EPDS < 10.5	43 (76.9%)	29 (70.7%)	
EPDS > 10.5	13 (23.1%)	12 (29.3%)	

EPDS = Edinburgh Postnatal Depression Scale.

**Table 3 jcm-11-05504-t003:** Bivariate analysis of secondary endpoints at day 3.

	Total	Lockdown Group (ld group)	Post Lockdown Group (pld group)	*p*
	(*n* = 264)	(*n* = 129)	(*n* = 135)	
Feelings of inability to care for the child	35/250 (14.0%)	29/121 (24.0%)	6/129 (4.7%)	<0.001
Experiences of early motherhood	*n* = 253	*n* = 124	*n* = 129	<0.001
Ruined	71 (28.1%)	69 (55.6%)	2 (1.6%)	
Neither wasted nor ideal	112 (44.3%)	55 (44.4%)	57 (44.2%)	
Ideal	70 (27.7%)	0	70 (54.3%)	
Apprehension of a gap with the partner at home	28/246 (11.4%)	21/119 (17.6%)	7/127 (5.5%)	0.050
Suffering expressed by the partner	69/240 (28.7%)	56/116 (48.3%)	13/124 (10.5%)	<0.001
Breastfeeding	*n* = 250	*n* = 122	*n* = 128	0.638
Easy breastfeeding	142 (56.8%)	73 (59.8%)	69 (53.9%)	
Difficult breastfeeding	62 (24.8%)	28(23.0%)	34 (26.6%)	
Artificial feeding	46 (18.4%)	21 (17.2%)	25 (19.5%)	
Consultation with psychologist desired	26/253 (10.3%)	16/123 (13.0%)	10/130 (7.7%)	0.236
Done	19/26 (73%)	12/16 (75.0%)	7/10 (70.0%)	0.273
Early release	78/246 (31.7%)	52/119 (43.7%)	26/127 (20.5%)	<0.001

**Table 4 jcm-11-05504-t004:** Univariate analysis of risk factors for postpartum depression across all groups.

	Odds Ratio [95% CI]	*p*
Confined group membership	0.7 [0.3–1.8]	0.501
Age		
<24	0.5 [0.0–6.1]	0.621
25–29	ref	
30–34	0.5 [0.2–1.6]	0.240
>34	0.4 [0.1–1.3]	0.119
Parity ≥ 1	0.2 [0.1–0.6]	0.003
History of violence during childhood		
psychological	6.4 [1.1–37.6]	0.040
physical	1.6 [0.1–18.6]	0.707
History of medical affection impacting pregnancy	1.2 [0.3–4.3]	0.778
Presence of psychiatric history	2.3 [0.5–11.0]	0.303
Presence of familial psychiatric history	0.4 [0.1–1.9]	0.236
Presence of a difficult obstetrical history	1.5 [0.3–8.8]	0.652
Precarious situation	6.1 [0.5–70.3]	0.148
Homeless patient	NA	NA
Presence of marital difficulties	NA	NA
Existence of a migratory pathway for the patient	0.5 [0.1–3.9]	0.472
Toxic use	0.8 [0.2–3.3]	0.794
Current unwanted pregnancy	0.5 [0.1–3.9]	0.472
Pregnancy difficult to achieve	1.4 [0.4–5.2]	0.613
Current pregnancy at risk	1.0 [0.4–2.8]	0.995
Presence of per-partum stress	4.5 [1.6–12.6]	0.004
Type of delivery		
Caesarean section	0.6 [0.2–2.0]	0.406
Instrumental delivery	1.1 [0.3–3.7]	0.863
Complications after delivery		
Preterm birth	3.0 [0.2–50.0]	0.444
Severe delivery hemorrhage > 1000 mL	1.5 [0.1–17.4]	0.746
Traumatic experience of childbirth	5.1 [1.4–17.8]	0.012
Unwanted absence of partner at the delivery	0.5 [0.1–4.0]	0.479
Mother/child separation at birth	0.9 [0.1–9.5]	0.961
Length of stay in the maternity ≥ 3 days	0.9 [0.3–2.3]	0.808

NA = not applicable due to small number of patients.

## Data Availability

The data that support the findings of this study are available from the corresponding author upon reasonable request.

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
