# Peer review of "Postpartum Depression after Maternal Isolation during the COVID-19 Pandemic: The MUMI-19 Study (Mothers Undergoing Mental Impact of COVID-19 Pandemic)"

_jcm, 2022, doi:10.3390/jcm11195504_

Round 1
Reviewer 1 Report
Thank you for the opportunity to review this manuscript. I noticed that the EPDS is referred to as the Edinburgh Postnatal Scale and it should be Edinburgh Postnatal Depression Scale. I would also like to see mention of and justification for how the cut-off score was selected for the analyses.
Author Response
Dear Reviewer,
Thank you for your reading and your comments.
We deeply appreciate the time and suggestions of the reviewer and have responded to his comments. The manuscript has been modified in accordance with his suggestions.
We hope that these changes in the revised manuscript now make our paper suitable for publication.
Best regards,
Vincent Dochez
Reviewer1:
Thank you for the opportunity to review this manuscript. I noticed that the EPDS is referred to as the Edinburgh Postnatal Scale and it should be Edinburgh Postnatal Depression Scale. I would also like to see mention of and justification for how the cut-off score was selected for the analyses.
Thank you for your synthesis and your comments.
We corrected “Edinburgh Postnatal Depression Scale” in the abstract and P4 L121.
P5 L129, we wrote: “In our study, we chose that a score of 13 could indicate a high risk of depression [14]. “
We have corrected this sentence by specifying the cut-off used: “We used the cut-off of 10.5 as in other studies in the literature as the primary endpoint. We also analyzed as secondary endpoint the cut-off of 13 to detect a severe characterized depressive episode”
Reviewer 2 Report
Reviewer Comments on Postpartum Depressiojn after maternal isolation during the COVID-19 pandemic: the MUMI-19 study (Mothers Undergoing Mental Impact of COVID-19 pandemic)
This is a study that aims to compare the prevalence of postpartum depression, the experience of the maternity stay, breastfeeding, experience upon return home, and mental wellbeing of the partner across groups of postpartum women who experienced lockdown at the beginning of their postpartum period and those who were not under lockdown. As the pandemic has presented unprecedented stressors for perinatal populations, continued research on understanding the ways in which the pandemic, and more specifically, isolation, has impacted and will impact mental health for pregnant and postpartum is essential. Further, even less is understood about how lockdown restrictions have impacted the partners of postpartum women. Thus, the potential significance of this manuscript is high and the distinctions across cohorts across the pandemic are also highly relevant. Most important in the revisions needed in this paper is a line edit of the manuscript to refine the language to reflect clear and specific English grammar as well as more scientifically accepted terms (i.e., replace “bad childbirth experience” in abstract with “traumatic/negative childbirth experience” or “artificial feeding” with “Formula or other feeding approach”).
INTRODUCTION:
1) The introduction is clear and succinct. Especially helpful are the specifications of what the lockdown regulations were in France, with which this reviewer was not familiar..
METHODS:
1) In the methods, starting on line 82, this reviewer was confused by the language as to which EPDS cutoff will be used. On line 82, it says that 10.5 “was used in our study” and then in lines 88-89, the authors write that “In our study, we chose that a sscore of 13 could indicate a high risk of depression.” It becomes clear later on in the manuscript what the authors mean by this, but that should be clearer in this section.
2) Starting on line 114, the authors describe their statistical methods and describe using Mann-Whitney- Wilcoxon tests to compare groups on continuous variables. It is unclear as to why they selected this test vs. a t-test. The reviewer can assume that the data did not meet requirements for parametric testing, but it would be helpful to be specific, as these samples are larger than many that use nonparametric tests. Other rationale for method selection are clear.
RESULTS:
1) The authors show in Figure 1 the compliance/attrition in their study. It is clear that there was a lot of attrition over time in this study, which is acceptable. However, it would be helpful, especially since the authors show little differences between groups on prevalence of PPD, that there were no differences in those who did not complete measures at follow-ups and those who did. These are simple analyses to run and would enrich the findings (i.e., were those who dropped out more depressed and therefore these results may reflect a self-selecting group of participants at follow-ups).
2) It is interesting that the ld women were more likely to report that restrictive measures brought challnenges to their early maternity experiences at day 30 and 60 as well, but there were no differences in their EPDS scores at these times. This may give some rationale that, despite not being reflected in reported symptoms of depression, these women still may need additional support that may be missed in postpartum screenings. This may be included in discussion if the authors see fit.
DISCUSSION:
1) Discussion is clear and succinct.
Author Response
Dear Reviewer,
Thank you for your reading and your comments.
We deeply appreciate the time and suggestions of the reviewer and have responded to his comments. The manuscript has been modified in accordance with his suggestions.
We hope that these changes in the revised manuscript now make our paper suitable for publication.
Best regards,
Vincent Dochez
Reviewer2:
This is a study that aims to compare the prevalence of postpartum depression, the experience of the maternity stay, breastfeeding, experience upon return home, and mental wellbeing of the partner across groups of postpartum women who experienced lockdown at the beginning of their postpartum period and those who were not under lockdown. As the pandemic has presented unprecedented stressors for perinatal populations, continued research on understanding the ways in which the pandemic, and more specifically, isolation, has impacted and will impact mental health for pregnant and postpartum is essential. Further, even less is understood about how lockdown restrictions have impacted the partners of postpartum women. Thus, the potential significance of this manuscript is high and the distinctions across cohorts across the pandemic are also highly relevant. Most important in the revisions needed in this paper is a line edit of the manuscript to refine the language to reflect clear and specific English grammar as well as more scientifically accepted terms (i.e., replace “bad childbirth experience” in abstract with “traumatic/negative childbirth experience” or “artificial feeding” with “Formula or other feeding approach”).
Thank you for your synthesis and your comments. We do believe that the consequences of lockdown restrictions are not yet sufficiently described and that it is essential that other studies relating to this lockdown be published. We have also corrected some terms and had the article proofread by a native English speaker.
INTRODUCTION:
1) The introduction is clear and succinct. Especially helpful are the specifications of what the lockdown regulations were in France, with which this reviewer was not familiar..
Thank you for your comments.
METHODS:
1) In the methods, starting on line 82, this reviewer was confused by the language as to which EPDS cutoff will be used. On line 82, it says that 10.5 “was used in our study” and then in lines 88-89, the authors write that “In our study, we chose that a score of 13 could indicate a high risk of depression.” It becomes clear later on in the manuscript what the authors mean by this, but that should be clearer in this section.
Indeed, we have corrected this sentence as suggested by reviewer 1 (P5 L129).
2) Starting on line 114, the authors describe their statistical methods and describe using Mann-Whitney- Wilcoxon tests to compare groups on continuous variables. It is unclear as to why they selected this test vs. a t-test. The reviewer can assume that the data did not meet requirements for parametric testing, but it would be helpful to be specific, as these samples are larger than many that use nonparametric tests. Other rationale for method selection are clear.
Indeed, we specified this in our manuscript: " comparisons between groups for continuous variables were made using the Mann-Whitney-Wilcoxon test which is a non-parametric test, rather than the t-test, because the data did not meet the requirements of parametric tests.”
RESULTS:
1) The authors show in Figure 1 the compliance/attrition in their study. It is clear that there was a lot of attrition over time in this study, which is acceptable. However, it would be helpful, especially since the authors show little differences between groups on prevalence of PPD, that there were no differences in those who did not complete measures at follow-ups and those who did. These are simple analyses to run and would enrich the findings (i.e., were those who dropped out more depressed and therefore these results may reflect a self-selecting group of participants at follow-ups).
Indeed, it is interesting to check the response rate according to the initial score at D3. For this reason we have inserted a new figure (figure 2) which we hope will meet your expectations.
2) It is interesting that the ld women were more likely to report that restrictive measures brought challnenges to their early maternity experiences at day 30 and 60 as well, but there were no differences in their EPDS scores at these times. This may give some rationale that, despite not being reflected in reported symptoms of depression, these women still may need additional support that may be missed in postpartum screenings. This may be included in discussion if the authors see fit.
This is a very good comment that we included in our discussion (P10 L260).
DISCUSSION:
1) Discussion is clear and succinct.
Thank you for your comments.